# A compact model of *Escherichia coli* core and biosynthetic metabolism

**Marco Corrao** [1], **Hai He** [2]*, **Wolfram Liebermeister**[3], **Elad Noor**[4], **Arren Bar-Even**[5☙]

1 Department of Engineering Science, University of Oxford, Oxford, United Kingdom, 2 Department of Biochemistry and Synthetic Metabolism, Max Planck Institute for Terrestrial Microbiology, Marburg, Germany, 3 Université Paris-Saclay, INRAE, MaIAGE, Jouy-en-Josas, France, 4 Department of Plant and Environmental Sciences, Weizmann Institute of Science, Rehovot, Israel, 5 Group of Systems and Synthetic Metabolism, Max Planck Institute of Molecular Plant Physiology, Potsdam-Golm, Germany

☙ Deceased.

\* hai.he@mpi-marburg.mpg.de

**Data availability statement:** The model, together with all relevant data, is available on

## Abstract

Metabolic models condense biochemical knowledge about organisms in a structured and standardised way. As large-scale network reconstructions are readily available for many organisms, genome-scale models are being widely used among modellers and engineers. However, these large models can be difficult to analyse and visualise, and occasionally generate predictions that are hard to interpret or even biologically unrealistic. Of the thousands of enzymatic reactions in a typical bacterial metabolism, only a few hundred form the metabolic pathways essential to produce energy carriers and biosynthetic precursors. These pathways carry relatively high flux, are central to maintaining and reproducing the cell, and provide precursors and energy to engineered metabolic pathways. Focusing on these central metabolic subsystems, we present *i*CH360, a manually curated medium-scale model of energy and biosynthesis metabolism for the well-studied bacterium *Escherichia coli* K-12 MG1655. The model is a sub-network of the most recent genome-scale reconstruction, *i*ML1515, and comes with an updated layer of database annotations and a range of metabolic maps for visualisation. We enriched the stoichiometric network with extensive biological information and quantitative data, including thermodynamic and kinetic constants, enhancing the scope and applicability of the model. In addition, we assess the properties of this model in comparison to its genome-scale parent and demonstrate the use of the network and supporting data in various scenarios, including enzyme-constrained flux balance analysis, elementary flux mode analysis, and thermodynamic analysis. Overall, this model holds the potential to become a reference medium-scale metabolic model for *E. coli*.

Github at https://github.com/marco-corrao/iCH360. The code and files required to reproduce all analyses in this manuscript are available on Github at https://github.com/marco-corrao/iCH360_paper and on Zenodo at https://doi.org/10.5281/zenodo.11092781. Additional analyses and comparisons with other models are also available in these repositories.

**Funding:** UK Biotechnology and Biological Sciences Research Council (UKRI-BBSRC, BB/T008784/1, M.C.). The funders had no role in study design, data collection and analysis, decision to publish, or preparation of the manuscript.

**Competing interests:** The authors have declared that no competing interests exist.

## Author summary

Metabolism is central to life, influencing microbial function in complex environments. Traditional genome-scale metabolic models offer broad coverage but often lack precision without extensive curation, while small-scale kinetic models provide accuracy but are limited in scope. Here, we introduce a novel intermediate-sized metabolic model of *Escherichia coli*, aiming to strike a balance between these extremes. Our "Goldilocks-sized" model is comprehensive enough to represent all central metabolic pathways yet small enough for thorough curation. It is richly annotated, highly interpretable, and includes various types of data already mapped to the model. We showcase possible uses of this data-enriched model by presenting a number of analysis and simulation methods, including calculations of thermodynamically feasible steady states with realistic enzyme allocation. By setting a new standard in model annotation and usability, we hope that our model will pave the way for more realistic, comprehensive, yet easy-to-use metabolic models for microbiology, systems biology, and biotechnology.

## Introduction

Metabolic models are a valuable tool for biologists and biotechnologists who want to elucidate and engineer cell metabolism [1–3]. In their simplest form, such models encode basic biochemical knowledge, such as network structure, reaction stoichiometries, or known reaction directionalities, in a structured and standardised format. However, the scope of these models is often wider, including information on catalysing enzymes and kinetic parameters, as well as annotations that link model elements to external databases. Rapid development of high-throughput experimental and computational pipelines has led to genome-scale metabolic network reconstructions now existing for a wide range of microorganisms [2,4]. One of them, *Escherichia coli*, has been the most studied prokaryotic organism and, as such, its metabolism has been the subject of extensive modelling efforts spanning over three decades [5–8]. In particular, for the common laboratory strain *E. coli* K-12 MG1655, the most recent genome-scale reconstruction, *i*ML1515, accounts for 1877 metabolites and 2712 reactions, mapped in detail to 1515 genes [8].

Genome-scale metabolic network models (GEMs) provide a comprehensive picture of cell metabolism, and constraint-based modelling algorithms that use these models have shown remarkable predictive power, for example, when predicting gene essentiality in bacteria [9]. However, working with such large models comes with some disadvantages. In the absence of sufficient constraining or parametrisation, simulations based on large networks can easily lead to biologically unrealistic solutions. For example, when designing and testing gene knockout strategies, genome-scale networks often wrongly predict unphysiological metabolic bypasses that must be manually inspected and filtered out [10,11] (see Table A in S1 Text for some examples). Another issue is that, owing to their size and complexity, the analysis of genome-scale networks is often limited to relatively simple modelling frameworks, such as flux balance analysis (FBA), that can only answer a limited range of questions. More complex methods, including the sampling of metabolic flux distributions [12], elementary flux mode (EFM) analysis [13], thermodynamics-based metabolic flux analysis [14], or kinetic modelling can be used to gain additional insight into the governing principles and constraints of microbial metabolism, but are difficult to apply to large models. Finally, genome-scale models are often hard to visualise comprehensively, which can make the interpretation of computed flux distributions cumbersome and unintuitive.

For all these reasons, small-scale models of *E. coli* metabolism are commonly used instead, both for strain design and for the development of novel modelling frameworks. Among these, the *E. coli* Core model (ECC) developed by Orth et al. [15] has been widely used in the literature. Although popular as an educational and benchmark tool, ECC is limited in scope: it lacks, among others, most biosynthesis pathways, which would be relevant to many metabolic engineering applications. This limitation was addressed by Hädicke and Klamt [16], who constructed a medium-scale model, *E. coli* Core 2 (ECC2), as a subnetwork of *i*JO1366, the most up-to-date GEM available at the time [7]. ECC2 was obtained through an algorithmic reduction [17] that iteratively prunes reactions from a template model while retaining user-specified structural and phenotypic features, such as the ability to grow under a defined set of conditions. However, to enforce the desired phenotypes, the algorithm relied only on steady-state stoichiometric modelling and did not account for other important factors, such as thermodynamics, kinetics, or regulatory effects, which are relevant under physiological conditions. Therefore, while the resulting submodel satisfies the stoichiometric constraints imposed by construction, further manual curation is often needed depending on the application at hand.

Here, we introduce *i*CH360 (named, according to convention, by the initials of the authors followed by the number of genes covered by the model), a manually curated "Goldilocks-sized" model of *E. coli* K-12 MG1655 energy and biosynthesis metabolism. The model was derived from the most recent genome-scale reconstruction (*i*ML1515 [8]) and includes all pathways required for energy production and for the biosynthesis of the main biomass building blocks, such as amino acids, nucleotides, and fatty acids, while the conversion of these precursors into more complex biomass components is described by a compact biomass-producing reaction. We extended the coverage of annotations that point to external databases from *i*ML1515, and built custom metabolic maps to facilitate the visualisation of the model and its subsystems. We complemented the stoichiometric network structure with a curated layer of biological information on catalytic function, protein complex composition, and small molecule regulation. Finally, we enriched the model with useful quantitative data, including thermodynamic and kinetic constants. Thanks to all these extra layers of information, our model can support a wide range of modelling methods beyond simple stoichiometric ones.

In the following, we present the model and demonstrate several use cases across various modelling scenarios, including enzyme-allocation predictions, EFM analysis, and thermodynamic analysis. The model is freely available in the standard formats SBML and JSON and can be used directly with popular metabolic modelling tools such as COBRApy [18].

## Results

### A compact model of *Escherichia coli* energy and biosynthesis metabolism

To assemble the *i*CH360 model, we started from the core metabolic reactions present in ECC and extended them with a curated set of pathways required for the biosynthesis of the main biomass building blocks, including the twenty amino acids, the five nucleotides, and both saturated and unsaturated fatty acids (Fig 1, Table 1, and Figs A–D in S1 Text). On the other hand, we deliberately did not include in our model the pathways required for the biosynthesis of complex biomass components and polymers, most degradation pathways, the pathways involved in the *de novo* biosynthesis of cofactors, and the reactions involved in the uptake of metals and ions. In addition, while not performing a comprehensive review of *i*ML1515, we applied a small number of corrections to the original reactions based on knowledge from the literature (Sect A in S1 Text).

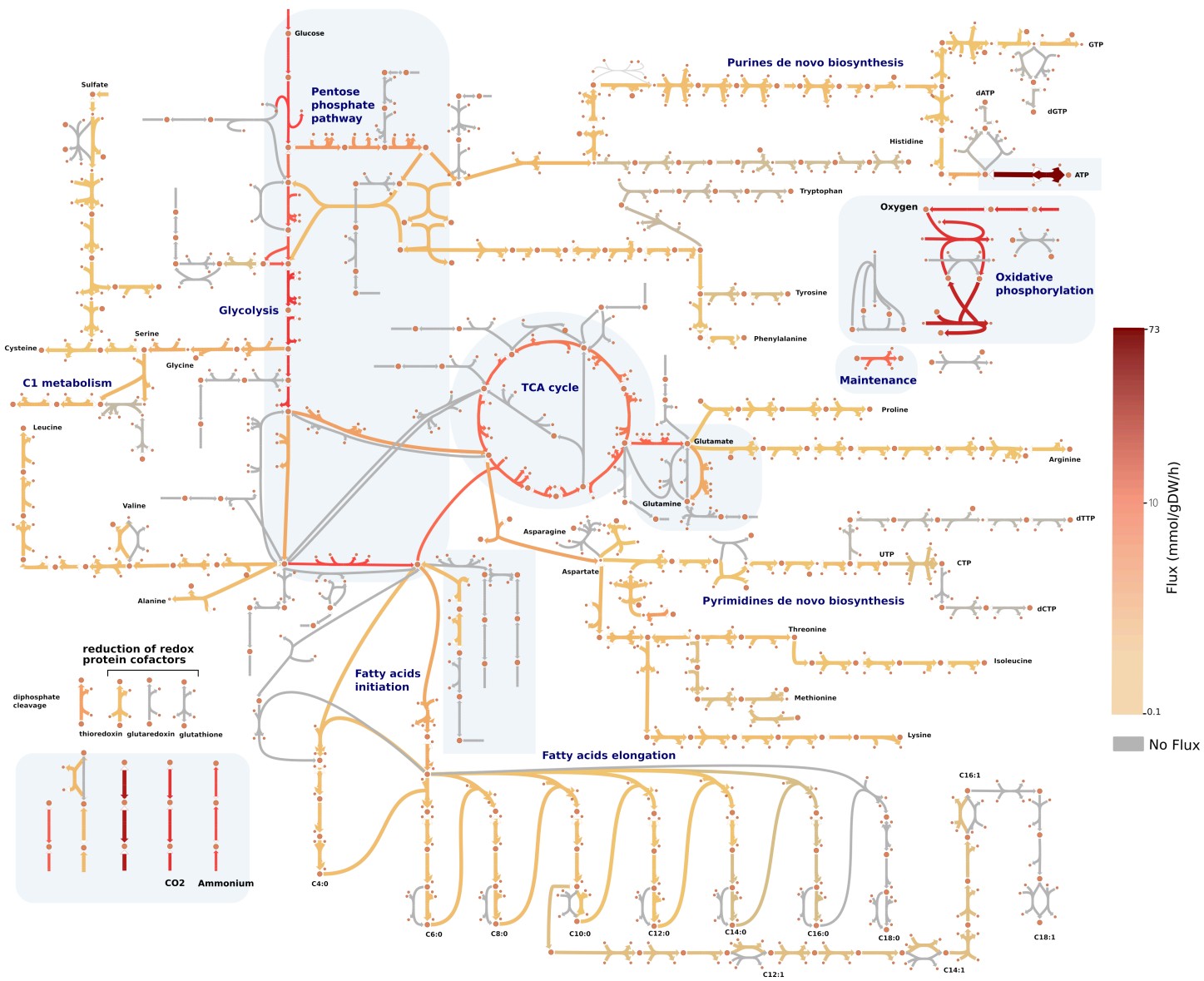

**Fig 1. Metabolic map of the *i*CH360 model.** The map was created with the metabolic visualisation tool Escher [19] and shows the metabolic subsystems included in the model. Shaded areas denote metabolic subsystems already present in the ECC model [15]. Reaction and metabolite names were omitted from the plot for clarity. Overlaid onto the map is a flux distribution for aerobic growth on glucose, computed via parsimonious FBA.

The final assembled model (Fig 1) comprises 304 compartment-specific metabolites (254 chemically unique compounds) and 323 metabolic reactions mapped to 360 genes, thus qualifying as a medium-scale model ranging between ECC and *i*ML1515 (Fig E in S1 Text). Although similar in scale, our model and ECC2 present a fundamental structural difference. ECC2 was built by systematically removing reactions from its genome-scale parent (*i*JO1366) [16]. Thus, its metabolic space spans the production of all compounds consumed in the *i*JO1336 biomass reaction. In contrast, the metabolic space of *i*CH360 only reaches biomass building blocks, without explicitly considering peripheral pathways such as the assembly of cell-membrane components, *de novo* synthesis of cofactors, and active transport of ions in

**Table 1.** The main metabolic subsystems covered by *i*CH360.

| Subsystem | Description | Metabolic map |
|---|---|---|
| Carbon uptake and transport | Reactions required for the uptake and assimilation of the following carbon sources: glucose, fructose, ribose, xylose, lactate, acetate, gluconate, pyruvate, glycerol, glycerate, succinate, alpha-ketoglutarate, malate | Fig 1 |
| Central carbon metabolism | Glycolysis, pentose phosphate pathway, pyruvate fermentation, TCA cycle, oxidative phosphorylation | Fig 1 |
| Amino acids biosynthesis | Biosynthesis of all 20 amino acids from core metabolism precursors | Fig A in S1 Text |
| Nucleotide biosynthesis | Biosynthesis of purine and pyrimidine nucleotides (and deoxynucleotides) from core and amino acid metabolism | Fig B in S1 Text |
| Fatty-acids biosynthesis | Biosynthesis of saturated and unsaturated fatty acids present in *E. coli* from acetyl-CoA | Fig C in S1 Text |
| C1 metabolism | One-carbon metabolism | Fig D in S1 Text |

the cell. To make the model comparable to its parent model *i*ML1515, we constructed an equivalent biomass reaction in which all biomass requirements not present in our model are summarised by an equivalent metabolic cost in terms of precursors, based on manually curated bioproduction routes (Table 2, Sect B in S1 Text, and File A in S1 Data).

## Range of metabolic conversions described by production envelopes

To check how its significantly smaller size and complexity affect the solution space of our model, we first looked at the maximum achievable biomass production flux under a range of growth conditions (Fig F in S1 Text). Across most conditions considered, the model achieves biomass fluxes comparable to those of its genome-scale parent (average relative difference of 2.3%, based on a carbon source uptake flux of 10 mmol/gDW/h and a cut-off on growth rate of 0.05 $h^{-1}$ to consider a model able to grow in a given condition). The main differences exist in anaerobic growth on fumarate, alpha-ketoglutarate (AKG), malate, and glycerol, where our model predicts zero growth, while the GEM achieves some (albeit small) biomass production rate. In practice, fermentation in these scenarios is biologically unrealistic (e.g. [20]). The reduced model is thus a good basis for metabolic simulation frameworks with few constraints, such as FBA, since the reduction procedure, at least in this test, limits the original solution space of the GEM to more physiologically relevant regions.

To further investigate the metabolic capabilities of our model, we looked at production envelopes (projections of the model solution space onto a smaller set of dimensions, also

**Table 2.** Biosynthesis pathways outside of the *i*CH360 model that were considered to construct a biomass reaction equivalent to the biomass reaction in the *i*ML1515 model. The right column shows the model identifiers of the main precursors present in *i*CH360. Note that only the main precursors are shown here, but the equivalent biomass reaction computed also accounts for any net production or consumption of metabolites in the reduced model.

| Pathway | Precursor in *i*CH360 |
|---|---|
| Biosynthesis of phosphatidylethanolamine (C16:0 and C16:1) | 3GP, PalmACP (C16:0), HdeACP (C16:1) |
| Biosynthesis of KDO2-Lipid-A | F6P, Ru5P, 3hmrsACP |
| Murein Biosynthesis | G3P, PEP, F6P, Ala |
| NAD/NADP de novo biosynthesis | Asp, DHAP |
| FAD de novo biosynthesis | GTP, Ru5P |
| CoA de novo biosynthesis | 3MOB, Asp |
| Active transport of ions | ATP |

known as phenotypic phase planes) describing production rates for biomass and a range of metabolites (Methods). Fig 2 shows the resulting envelopes for aerobic growth on glucose. The reduced model has production envelopes comparable to *i*ML1515, except for the production of acetate, where the genome-scale model can achieve considerably higher yield (moles of acetate produced per mole of carbon source), both aerobically and anaerobically.

In order to understand the cause of these differences, we investigated optimal acetate production routes in both models using FBA (Sect C in S1 Text). Our analysis traced this discrepancy in the ability of the genome-scale model to achieve higher production yields for acetyl-CoA (in the aerobic case) and pyruvate (in the anaerobic case), using additional degradation routes not included in our model, and even fixing carbon from external $CO_2$ (Sect C and Fig G in S1 Text). In light of this analysis, these higher maximal yields achievable by *i*ML1515 for acetyl-CoA and pyruvate appear unrealistic. Hence, in agreement with what was previously done with ECC2 [16], we chose not to include in *i*CH360 any additional reactions that would allow for higher acetate yields. A similar pattern was found when comparing production envelopes between the two models under different growth conditions (Fig H in S1 Text).

Production envelopes generated with *i*CH360 were also comparable, albeit not identical, to those computed on other existing medium-scale models, namely ECC and ECC2 (Fig I in S1 Text). Particularly, the maximal yields achievable for each product were nearly identical for the products considered, both aerobically and anaerobically. Some differences between the envelopes can be observed for solutions with higher predicted growth rates (on the right-hand

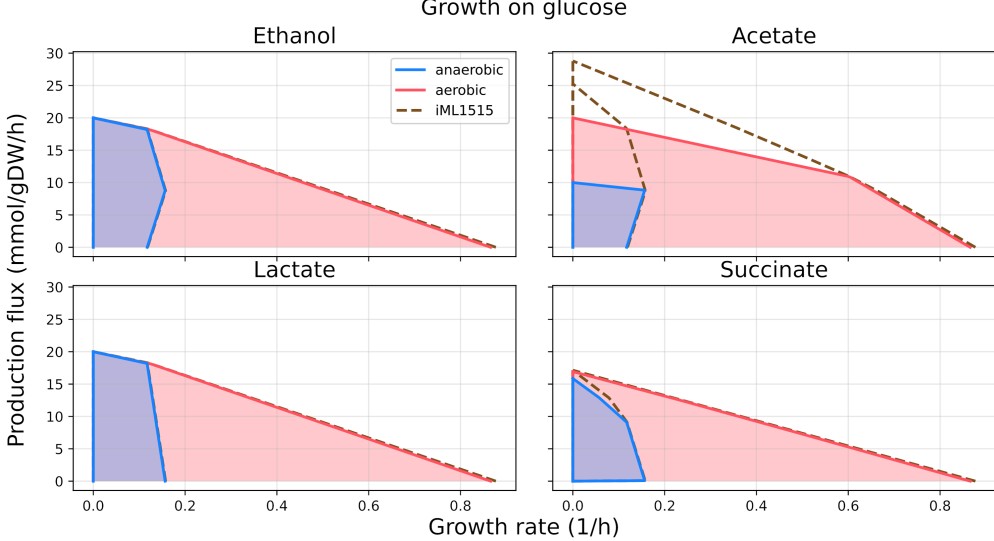

**Fig 2. The *i*CH360 model shows similar, but more realistic metabolic capabilities than *i*ML1515.** Considering glucose as a feedstock and studying ethanol, lactate, and succinate production, a production envelope analysis yields similar results in the two models (note that the dashed line representing the production envelope of *i*ML1515 is sometimes hidden behind the coloured lines). Growth rate and production fluxes were computed by limiting the glucose uptake rate to a maximum of 10 mmol/gDW/h, so that the production yield (in moles of product per mole of carbon source) can be obtained by dividing the production flux (y-axis) by 10. In the scenario of acetate production (top right panel) *i*CH360 avoids an unrealistically high production flux [16] as predicted by *i*ML1515. An extended set of production envelope comparisons between the two models is available online in the code repository supporting this manuscript.

side of the envelopes). However, since the three models are equipped with different biomass reactions sourced from different parental models (and hence predict different biomass yields), these differences are expected.

## Connecting reactions to their catalysing enzymes and the enzymes' protein components

Metabolic models often contain annotations that connect model elements to entries in biological databases, such as BioCyc [21], KEGG [22] and MetaNetX [23]. However, even for the subset of reactions included in *i*CH360, the annotations present in *i*ML1515 were incomplete and, in part, outdated. To fill these gaps, we extended and corrected the original annotations through a mixture of automated querying and manual curation (Fig 3A). Notably, the annotations pointing to the BioCyc knowledgebase [21] are nearly complete: Out of 321 enzymatic reactions in the model, 317 are mapped to BioCyc with a single ID (for the remaining four unannotated reactions, involved in the biosynthesis of unsaturated fatty acids, a match in the database could not be found for the specific use of NADPH as a redox cofactor). Further, nearly all of these BioCyc annotations (315 / 317) are in the ECOLI namespace and therefore point to the organism-specific EcoCyc database, a widely used and extensive reference for *E. coli* molecular biology [24,25]. The remaining two reactions map instead to the broader MetaCyc database, also part of the BioCyc ecosystem, via the META namespace. Additionally, we added 13 reaction annotations mapping to the KEGG database and found 134 deprecated annotations pointing to the MetaNetX database, which were consequently updated with the most up-to-date IDs.

Using the extensive mapping to the EcoCyc database, we parsed, assembled, and manually curated a knowledge graph that enhances the stoichiometric model with a detailed layer of information about enzymes, polypeptides, and genes related to the network (Methods). This data structure takes the form of a weighted graph, where nodes represent biological entities (reactions, proteins, genes, or compounds) and edges represent (potentially quantitative) functional relationships between them (Methods, Fig 3B, and Tables B–C in S1 Text), such as catalysis, regulation, protein modification, and protein subunit composition. This graph contains the information collected in a unified form, allowing users to perform a number of tasks that occur in metabolic modelling applications. For example, by explicitly mapping reactions to their catalysing enzymes rather than to the associated genes (which may additionally include protein activators or cofactors), it simplifies the definition of meaningful enzyme capacity constraints in the model. Similarly, since the graph topology implicitly defines associations between reactions and genes, Boolean gene-protein-reaction (GPR) rules needed for *in silico* knockout studies can be generated that account for catalytic and non-catalytic requirements for each reaction (Sect D in S1 Text). Crucially, these GPR rules can be regenerated as needed whenever the graph is updated with new nodes or edges. Finally, we used the graph to estimate the abundance of protein complexes included in the model from the measured polypeptide abundances using a simple automated procedure (Sect E in S1 Text).

Through this annotation graph, 318 metabolic reactions in the model are linked to 289 catalysing enzymes, with more than 25% of the reactions being catalysed by multiple enzymes (isozymes). Such enzymatic redundancy plays an important role, for example, when designing metabolic engineering strategies to prevent flux through a pathway. However, the different isozymes of a reaction need not all be equally important, and treating them as completely equivalent may generate inaccuracies in some phenotypic predictions. For example, while phosphofructokinase activity in *E. coli* (reaction PFK) is known to be carried out by

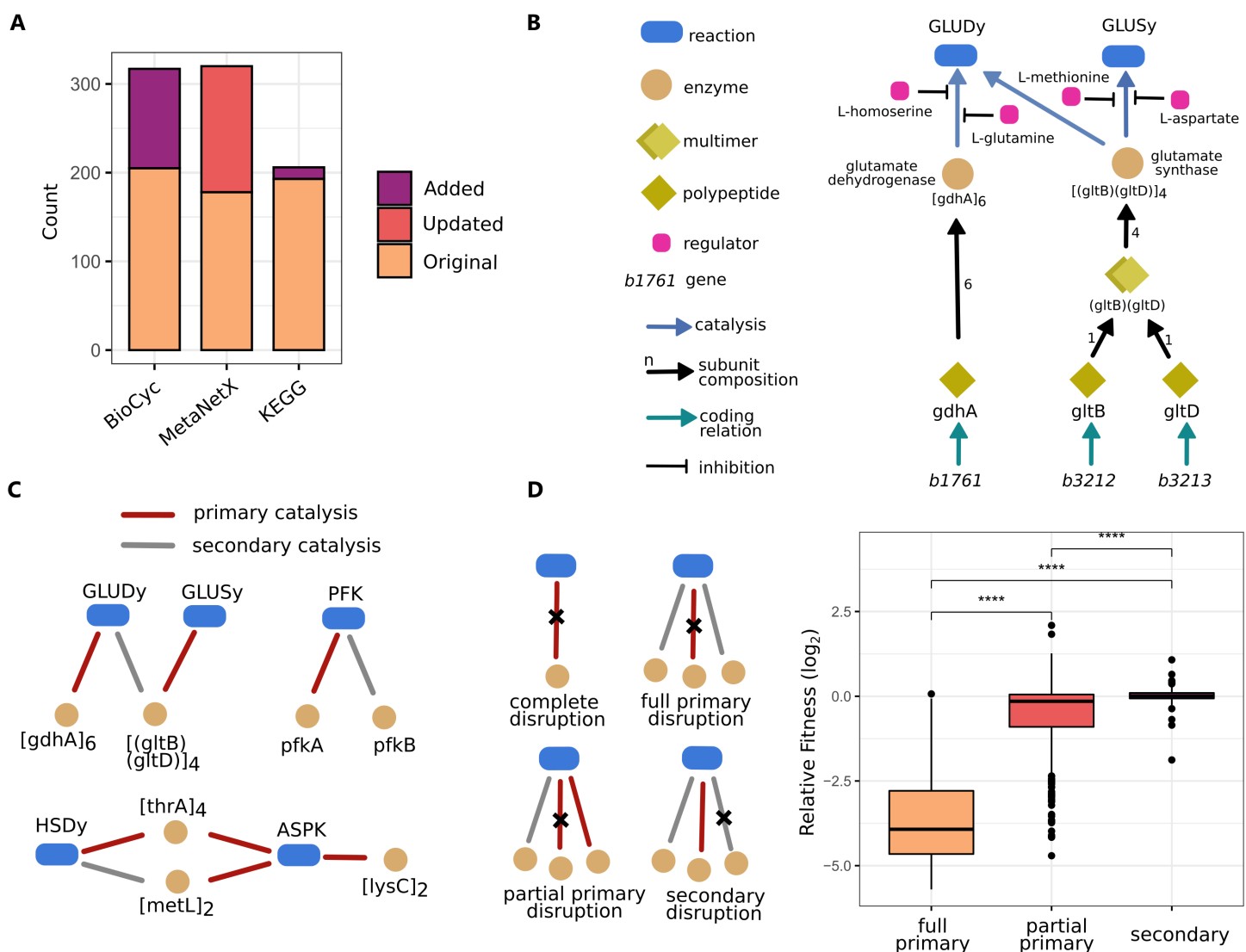

**Fig 3. Layers of annotation and biological knowledge supporting the stoichiometric model in *i*CH360. A**: Annotations for the model reactions point to the BioCyc, MetaNetX, and KEGG databases. Bars show the number of annotations, highlighting the share of annotations that were added to or updated from the parent model *i*ML1515. **B**: Some of the biological knowledge parsed from EcoCyc (and manually curated) included in the model-supporting functional annotation graph. The graph captures catalytic relationships between reactions and enzymes, protein subunit compositions, protein-gene mappings, and small-molecule regulation interactions, among others. Shown here as an example are the branches of the graph corresponding to the Glutamate Dehydrogenase (GLUDy) and Glutamate Synthase (GLUSy) reactions. **C**: Examples of catalytic relationships functionally annotated as either primary or secondary in the graph. Note that all catalytic relationships were classified as primary by default, unless sufficient evidence was found to annotate them as secondary. **D**: Functional annotation of catalytic edges as primary or secondary can be used to improve phenotypic predictions. Left: Classification of catalytic edge disruptions in the network resulting from simulated knockout of genes associated with essential reactions in the model across 9 growth conditions (see text for a description of each disruption class). Right: Comparison of predicted disruption outcomes against a large dataset of mutant fitness data [27] shows that the different types of disruption tend to lead to significantly different fitness changes. Whiskers in the box plot denote the range of data located 1.5 times above and below the interquartile range. Black dots represent data points lying outside this range.

two isozymes, *pfkA* and *pfkB*, the latter is known to be responsible only for minor activity under normal physiological conditions [26]. If these differences are not taken into account, metabolic models can overpredict redundancy in the network, for example, when predicting phenotypes after gene knockouts [8].

To address this issue and make the model usable in a wider range of modelling scenarios, we classified the catalytic edges in the graph as either primary or secondary catalysis (Methods, Fig 3C). The catalytic relationship between a reaction and an enzyme was annotated as secondary whenever the enzyme, according to experiments, accounts only for negligible activity for the reaction in the wild-type strain. As experimental evidence, we considered *in vitro* and *in vivo* complementation studies. Through this curation process, a total of 72 catalytic edges were functionally annotated as secondary.

To test how this functional annotation can support phenotypic predictions, we identified essential reactions across a range of growth conditions as those whose removal from the model abolishes growth. For each such reaction-condition pair, we identified all catalysing genes via our knowledge graph. We then performed simulated knockouts for each gene, propagated the disruption through the graph using Boolean GPR logic, and assessed the impact on the reaction(s). We classified the outcome of each knockout as: (i) complete disruption, if a reaction loses all of its catalytic edges; (ii) full primary disruption, if a reaction loses all of its primary catalysis edges, but secondary ones are left; (iii) partial primary disruption, if the reaction loses some, but not all, of its primary catalysis edges; or (iv) secondary disruption, if some or all secondary edges are disrupted, but none of the primary ones. These gene-condition-disruption tuples were then mapped to corresponding experimental fitness values, obtained via competitive fitness assays, from Price et al. (2018) [27].

Based on this analysis, we found that disruptions of the primary edges as a result of a knockout were significantly associated with greater fitness losses than disruptions of secondary edges (Wilcoxon rank-sum test, $p<10^{-6}$, Fig 3D). In addition, primary disruptions that were only partial were associated with more contained fitness losses. Finally, minor fitness gains could be identified when mutants were associated with secondary and partial primary disruptions, but not when complete primary disruptions occurred. We did not find any significant differences between complete disruptions and full primary disruptions (Fig J in S1 Text), supporting the idea that catalytic relationships annotated as secondary are unlikely to be strong enough to stand-in for disrupted primary ones under normal physiological conditions. While this annotation is, at this stage, of qualitative nature, future incorporation of quantitative distinctions between different catalytic relationships would allow for its direct integration into simulations of the model.

## Enzyme-constrained flux balance analysis with EC-*i*CH360

To make *i*CH360 applicable for enzyme-constrained flux simulations, we constructed a version of the model containing all necessary extra information, which we denote as EC-*i*CH360. We constructed EC-*i*CH360 in the sMOMENT [28] format (Methods). The sMOMENT framework is inherently simple and generates the same solution space as more complex model formats, such as GECKO, unless reaction-specific capacity constraints are specified in the latter [28,29]. Since it requires unique reaction-enzyme mappings, we used the knowledge graph to remove all secondary catalytic relationships in the model.

We first parametrised the model by defining a flux-specific enzyme cost for each reaction (in units of grams of enzyme per unit flux), using as parameter values the estimated *in vivo* turnover numbers from Heckmann et al. (2020) [30]. Using this enzyme-constrained model, we then predicted enzyme abundances and compared them to experimental enzyme abundances, estimated from a dataset of proteomic measurements for aerobic growth on eight different carbon sources [31] (Methods). This analysis led to predictions with root mean squared

error (RMSE, computed for $\log_{10}$-transformed enzyme abundances) ranging from 0.53 to 0.62 (Fig K in S1 Text). To assess the nature of residuals between measurements and predictions, we investigated the geometric mean of enzyme abundances across all eight conditions (Fig K in S1 Text, bottom right panel). If the mismatch between measurements and predictions were due to each enzyme operating at different saturation levels in each condition, one would expect that averaging would reduce these differences. However, averaging the predicted and measured enzyme abundances, respectively, across conditions did not significantly improve the RMSE, indicating that the abundances of individual enzymes were systematically over- or under-predicted between conditions.

To increase the predictive capacity of the model, we adjusted the turnover numbers by fitting them to experimental measurements through a custom heuristic (Methods, Sect F in S1 Text). By simultaneously fitting all available conditions with a single set of parameters, we ensured that our adjustment procedure is robust to condition-specific biases. Furthermore, by introducing regularisation within our adjustment scheme, we penalised large deviations of parameters from the original data set, thereby increasing the robustness of the procedure to overfitting. Our adjusted parameter set shows a mean absolute deviation (computed for $\log_{10}$-transformed turnover values) from the original parameter set of $\approx 0.22$ (Fig L in S1 Text) and results in significantly better agreement with experimental measurements across conditions (Fig 4A). Further, mean enzyme abundances across conditions are very well predicted with the adjusted parameter set (Fig 4B), implying that residuals between measurements and predictions are now to be attributed to variability in enzyme saturation across conditions (which simple frameworks such as enzyme-constrained FBA, which use a constant enzyme cost per unit flux, cannot account for), rather than systematic over- or under-predictions. Thus, each adjusted turnover parameter can be thought of as a "typical" apparent $k_{\text{cat}}$ value, incorporating average saturation trends for an enzyme across growth conditions.

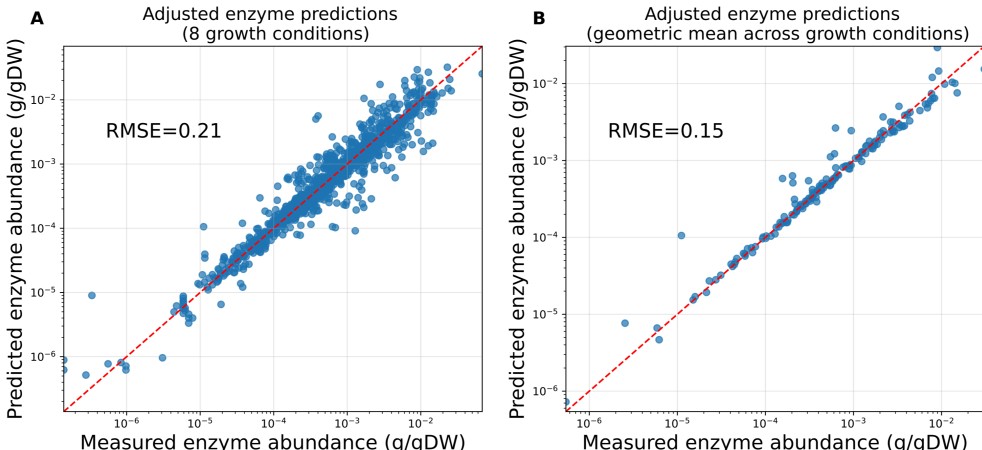

**Fig 4. Enzyme allocation predictions obtained with the model variant EC-*i*CH360 after adjusting the turnover parameters. A**: Predicted vs measured enzyme abundances for aerobic growth on eight different carbon sources. Each data point represents an enzyme-condition pair. A total of 325 data points corresponding to zero predictions (enzymes associated with zero-flux in the enzyme-constrained FBA solution for a given condition) were omitted from the plot. **B**: Geometric mean across conditions of predicted vs measured enzyme abundances. For each enzyme, the geometric mean was computed across the conditions with non-zero predicted abundance. A total of 27 data points, corresponding to enzymes with zero predictions across all conditions, were omitted from the plot.

To further confirm this aspect, we performed a leave-one-out cross-validation analysis (Methods), where each condition was, in turn, excluded from the model fitting dataset but used to evaluate predictions. Results (Fig N in S1 Text) show that the predictions of enzyme abundances in a given condition are considerably improved even when data from that condition is not used for fitting, confirming that our parameter fitting heuristics captures global trends and not condition-specific effects. Training the model on the full dataset, as we did to compute the final parameter set, further improves predictions, even if to a lesser extent. This can be explained by noting that, by design, our procedure cannot adjust the turnover parameters for enzymes associated with zero flux in the reference flux distribution used for fitting (Sect F in S1 Text). Hence, including all conditions in the training set allows for the turnover number of highly condition-specific enzymes (those associated with non-zero flux in only one of the reference flux distributions used by the procedure) to also be adjusted, further improving the overall prediction metrics.

## Elementary flux modes in the reduced model variant *i*CH360$_{red}$

Despite the small size of *i*CH360, we found the explicit enumeration of its elementary flux modes (EFMs) to be intractable. This is not necessarily surprising since the EFM count is crucially dependent on the topology of a metabolic network rather than its sheer size. Metabolic networks can possess different types of redundancy, such as the presence of alternative pathways for the production of the same metabolite, the use of alternative cofactors for the same catalytic step in a pathway, or the presence of alternative transporters for the uptake/excretion of a compound. Although knowledge of these redundancies is often valuable, including them in the model can increase the number of EFMs exponentially, hampering or even preventing EFM-based analyses.

To address this issue, we identified and removed a small set of alternative metabolic routes in *i*CH360, using available information from the literature whenever possible to ensure that the most physiologically relevant alternative was maintained (Table D in S1 Text). This resulted in a metabolic submodel of *i*CH360, a model variant that we denote by *i*CH360$_{red}$. *i*CH360$_{red}$ contains 305 metabolic reactions (18 less than *i*CH360) and shows the same production envelopes as its parent model for a number of metabolites of interest (Fig O in S1 Text). While the number of EFMs in *i*CH360$_{red}$ is still relatively large ($\approx$ 13.5 millions for aerobic growth on glucose, see Table E in S1 Text), it is not prohibitive for most types of EFM-based analysis, and their explicit enumeration does not require high-performance computing (Methods).

We used the EFMs of *i*CH360$_{red}$ to study the possible combinations of achievable growth rates and yields in the network [13]. To this end, we considered growth on glucose as a scenario and determined, for each EFM from *i*CH360$_{red}$, its yield, computed as the ratio of biomass flux and glucose uptake, and its achievable cell growth rate, which we estimated based on the enzyme costs defined for the enzyme-constrained model (Methods). Based on this analysis, we identified a front of Pareto-optimal EFMs, along which any increase in the growth rate will necessarily lead to a reduction in yield (Fig 5). Along the Pareto front, we observe a transition from a purely respiratory mode at maximum yield (Fig P in S1 Text) to a mixed respiratory-fermentative mode at maximum growth (Fig Q in S1 Text). Quantitatively, the extent of this trade-off was rather modest: the EFM with maximal yield reaches almost the maximal growth rate, so only minor gains in growth rate can be achieved by using other, fermentative modes along the Pareto front. However, it is worth noting that this analysis was performed using a simple capacity-based enzyme cost function, which ignores metabolite

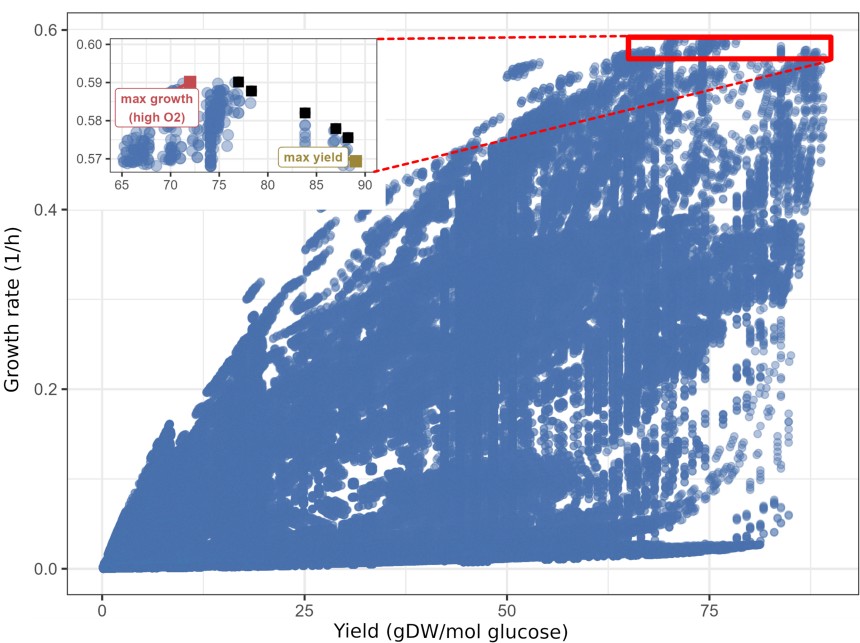

**Fig 5. Growth rates and biomass yields achieved by different elementary flux modes of *i*CH360$_{red}$ for growth on glucose.** The inset on the top left (corresponding to the area of the plot enclosed by the red rectangle) highlights the front of Pareto-optimal EFMs (squares), with the maximum-growth and maximum yield modes lying at the extremes of the front. The growth rate of each mode was estimated by assuming that the metabolic enzymes in the model occupy, by mass, a constant fraction of the cell's dry weight (see Methods).

concentrations by assuming a constant enzyme cost per unit flux for each reaction. Repeating the analysis with a more complete enzyme-cost function, such as one that accounts for variable thermodynamic driving force and enzyme saturation, could help elucidate the nature of this trade-off [32]. To demonstrate that the shape and size of the Pareto front depend strongly on growth conditions, we also simulated an environment with very low oxygen levels. We implemented this by increasing the flux-specific enzyme cost of the oxygen-dependent reaction in the respiratory chain, equivalent to assuming a lower enzyme efficiency due to a lower oxygen level (Methods). Results (Fig R in S1 Text) show a much broader front of Pareto optimal EFMs, indicating that the nature of the observed trade-off is indeed condition-dependent. Notably, a similar dependence of the Pareto front on extracellular oxygen availability has been previously observed in a small-scale model of *E. coli* core metabolism [13].

## Saturation FBA and modelling of overflow metabolism

In order to study the effect of external conditions on optimal metabolic strategies in more detail, we used another framework that does not require an enumeration of EFMs and allows for additional flux bounds, for example, to impose a minimum ATP consumption rate for cell maintenance. The saturation FBA (satFBA) framework [33] is a variant of enzyme-constrained FBA, wherein a fixed enzyme cost per flux is assumed for all metabolic reactions in a model, except for the substrate transporter, for which a complete kinetic law is used (Methods). Since the external substrate concentration is a simple parameter, screening this concentration is equivalent to screening the values of the transporter efficiency. Here we used

satFBA to simulate how the growth-maximising solution of the network varies in response to changing extracellular glucose concentration. By solving the satFBA problem for a range of glucose concentrations, we predicted the dependence of the cell's growth rate on substrate concentration, resulting in the typically observed Monod curve (Fig 6A). If the problem does not contain any further flux bounds (so that the magnitude of fluxes at the optimum is solely limited by the maximum enzyme availability) the solution of satFBA problems will be an elementary flux mode [33,34]. Hence, in this case, we can use satFBA to explore how a cell should switch across elementary modes as a function of the growth environment.

At low glucose concentrations, the glucose transporter operates at low saturation, and glucose uptake is enzymatically expensive, leading to a high-yield, purely respiratory metabolic mode at the optimum (Fig 6B and C). As substrate availability is increased, the cost of substrate uptake decreases, and higher growth rates are achieved by switching to lower yield, acetate-secreting modes [33]. Since the yield of a flux distribution is, by definition, constant along an elementary flux mode, the yield varies in a step-like manner as a function of external glucose concentration (Fig 6C, inset), where each jump represents a change of optimal mode. The satFBA formalism can also be used with additional flux bounds. For example, if a positive lower bound on ATP hydrolysis is added as a maintenance requirement, optimal solutions to the satFBA problem (Fig S in S1 Text) will no longer be elementary modes, and the yield of the optimal solution no longer follows a piecewise constant profile (Panel C of Fig S in S1 Text).

## Equilibrium constants, thermodynamic forces, and thermodynamically feasible states

Living systems operate outside of thermodynamic equilibrium, and thermodynamics places strong constraints on the operation of metabolic systems. In any metabolic state, the flux directions must follow the signs of thermodynamic forces, which depend on metabolite concentrations and equilibrium constants. To provide these constants as parts of our model, we used the component contribution framework [35] to estimate the standard Gibbs free energy ($\Delta_r G'^\circ$) of each reaction (Methods). These estimates account for compartment-specific

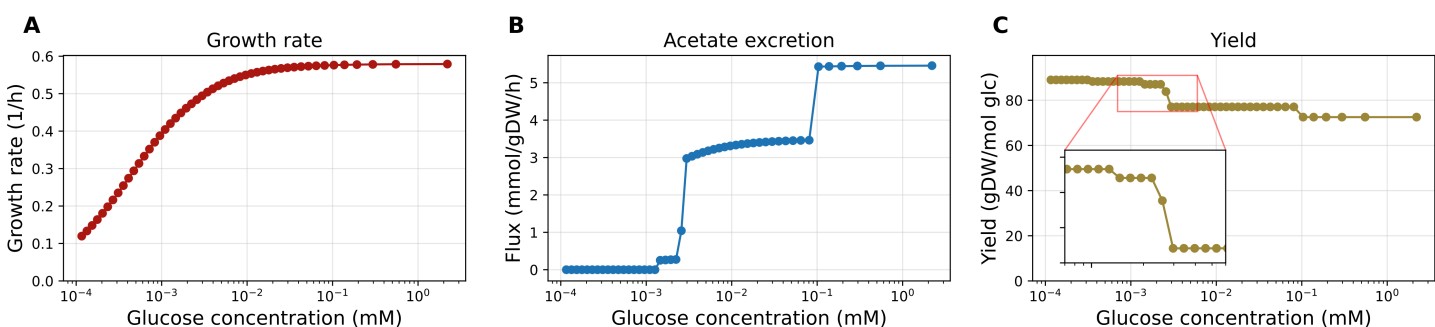

**Fig 6. Saturation FBA enables the exploration of the optimal switching across elementary modes as a function of the growth environment. A**: satFBA predictions for the growth rate as a function of external glucose concentration, showing a typical Monod curve. Note that satFBA computes the cell's growth rate by assuming a fixed total enzyme mass budget while varying the saturation of the substrate transporter as a function of external substrate concentration. Importantly, although the curve is continuous and smooth, it comprises many smaller sections, each dominated by a different elementary mode. **B**: satFBA predictions for the acetate excretion flux, showing progressively higher use of fermentative metabolism in the optimal solution as external glucose availability increases. **C**: The biomass yield of the optimal satFBA solution (in gDW/mol glucose) progressively decreases in a step-like manner as external glucose availability increases. Each jump represents a switch in the optimal elementary flux mode.

chemical environment parameters, such as pH, pMg, and ionic strength, and were corrected to account for protons and charge translocation in multi-compartment reactions.

The resulting parameter set covers the vast majority of the model (over 97% of metabolic reactions, with the remaining not covered by the component contribution database used here) and accounts for the uncertainty in the estimates through a multivariate covariance matrix. Accounting for the correlations between different $\Delta_r G'^\circ$ values becomes important when imposing thermodynamic constraints on the model. For example, the fatty acid biosynthesis subsystem in the model consists of repeated elongation cycles, where a short sequence of chemical transformations is repeatedly performed on a growing carbon chain. As a result, even if the $\Delta_r G'^\circ$ for each reaction in the pathway is known with some uncertainty, this uncertainty is tightly correlated across the reactions, which constrains the set of achievable thermodynamic states in the network.

Using this set of thermodynamic constants, we first tested whether some typical flux distributions obtained from the model are thermodynamically feasible under realistic metabolite concentration ranges. To this end, we considered flux distributions generated by parsimonious Flux Balance Analysis (pFBA) across 12 growth conditions and computed their max-min driving force (MDF) [36], accounting for uncertainty in the estimates (Methods). We found a positive MDF for each of the flux distributions, indicating that all pFBA solutions tested are thermodynamically feasible. Notably, we found that the computed MDF values cluster very clearly in three groups, corresponding to aerobic growth on glycolytic substrates (high MDF), aerobic growth on gluconeogenic substrates (medium MDF), and anaerobic conditions (low MDF, Fig 7A).

Having confirmed that our reference FBA-derived flux distributions are thermodynamically feasible, we then employed an alternative flux prediction method that ensures thermodynamic feasibility by construction. The probabilistic metabolic optimisation (PMO) framework [37] uses a mixed-integer quadratic programming approach to compute a set of

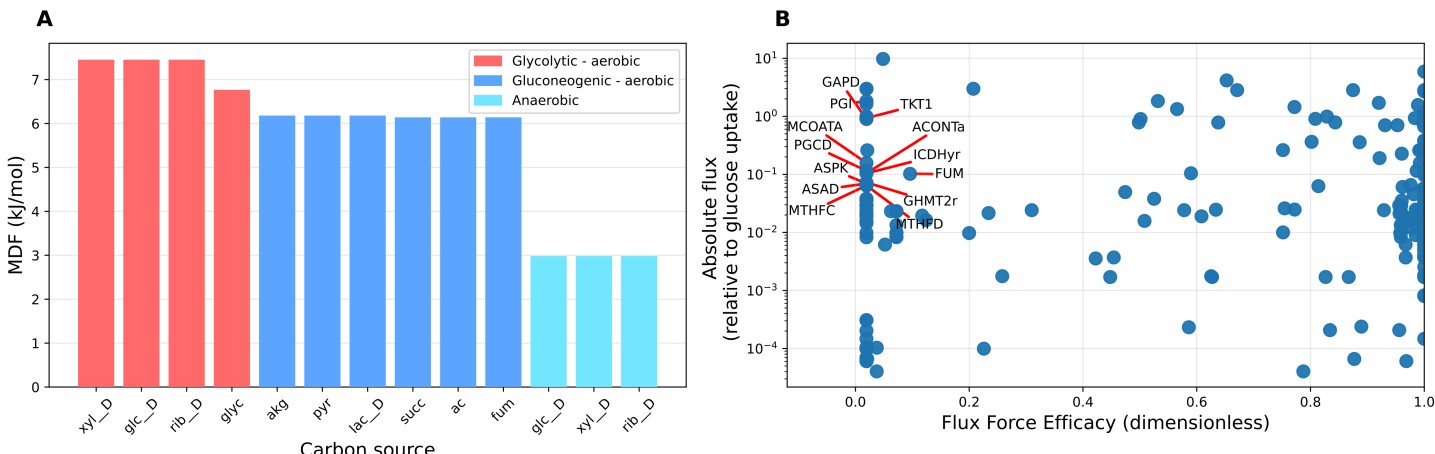

**Fig 7. Thermodynamic analysis of the model via the curated thermodynamic parameter set. A** Probabilistic max-min driving force (MDF) analysis of flux distributions obtained by parsimonious flux balance analysis for a total of 12 growth conditions. All flux distributions tested have a positive MDF, implying they are thermodynamically feasible under physiologically reasonable metabolite concentration ranges. The computed MDF values cluster in three groups, corresponding to glycolytic aerobic, gluconeogenic aerobic, and anaerobic growth conditions. **B**: Fluxes relative to the glucose uptake flux (EX_glc__D_e) and flux-force efficacies computed by probabilistic metabolic optimisation (PMO). The labelled data points represent examples of reactions (excluding transport and spontaneous reactions) with low predicted flux-force efficacy (here, below 20%), but carrying high relative flux in the optimal solution (here, more than 5% of the glucose uptake flux). xyl__D: D-xylose; glc__D: D-glucose; rib__D: D-ribose; glyc: glycerol; akg: alpha-keto-glutarate; ac: acetate; pyr: pyruvate; lac__D: D-lactate; succ: succinate. fum: fumarate.

fluxes, metabolite concentrations, and reaction driving forces that is probabilistically most in agreement with experimentally measured metabolite concentrations (Methods). Using this framework, we computed a maximum-likelihood thermodynamic state for the model, simulating aerobic growth on glucose. In the thermodynamic state computed by PMO, we found that all metabolite concentrations lie within physiologically reasonable ranges (1 $\mu$M – 1 mM). In addition, all "anomalous concentrations" identified by the framework (metabolite concentrations lying more than one standard deviation away from the mean experimental value) had been identified and explained previously [37]. Whenever substrate channelling (the process in which the product of an enzyme is directly passed to the active site of another enzyme, without release in the bulk phase) is known to happen, these thermodynamic anomalies can be addressed by lumping together the relevant reactions, resulting in the summation of their $\Delta_r G'^\circ$ (see Supplementary Information in [37] for a comprehensive analysis of this aspect).

We used the PMO-derived thermodynamic state to identify candidate bottlenecks in terms of enzyme demand across the network. To this end, we first computed the flux-force efficacy of each reaction in the thermodynamic state [36] (Methods). The flux-force efficacy is a unitless quantity, ranging from 0 to 1, denoting the ratio between net flux (forward minus backward flux) and total flux (forward plus backward flux) of a reaction. Reactions operating at low flux-force efficacy have a lower net flux due to two reasons: (1) the forward flux is lower in absolute terms, and (2) the higher backward flux is counter-productive and subtracts from the forward flux. Therefore, to achieve a given required net flux, the cell has to invest more resources in maintaining a higher enzyme level. To identify potential thermodynamic bottlenecks that lead to high enzyme costs in specific reactions, we therefore screened all reactions for their predicted flux-force efficacy and flux (Fig 7B) and identified those predicted to operate at low efficacy while carrying significant flux (Fig 7B, labelled points).

To assess the extent to which the thermodynamic operating states predicted by PMO are predictive of enzyme investment, we split reactions into two groups based on whether their predicted flux-force efficacy sits above or below 50%, and compared the distributions of measured enzyme abundances between the two groups (Methods, Fig T in S1 Text). While there are many determinants of enzyme abundance beyond thermodynamics, including enzyme turnover, affinity, and regulation, we observed a significant difference between the two distributions ($p<0.001$, two-sided Wilcoxon rank-sum test), with the low efficacy group having approximately a 3-fold higher median enzyme abundance than the high efficacy group. Establishing a causal link can be very difficult, but we can speculate that this negative correlation might be explained by metabolism evolving to compensate for the low efficacy of some reactions with a higher expression level of their catalysing enzyme [36,38].

## Discussion

Here we presented *i*CH360, a medium-scale metabolic model of *E. coli* covering central and biosynthesis metabolism, together with the associated data and metabolic maps and results from several analysed use cases. While designed for the laboratory strain K-12 MG1655, its restricted metabolic coverage, focused on central pathways, makes the model applicable to other *E. coli* strains. Similarly to previously constructed core models [15,16], this model trades metabolic coverage for usability, interpretability, and ease of visualisation. It is well suited whenever a relatively small, highly curated network is desired, when computationally demanding analyses are to be performed, or as an educational tool in the field of metabolic modelling. When comparing some key properties of this model with those of its parent genome-scale model, we observed only small differences in the achievable biomass and

product yields across a range of growth conditions, validating that, despite its contained size, the model captures the most salient metabolic features of the genome-scale network.

Further, we showed that the use of a well-curated, smaller-scale model can, in some cases, even correct unrealistic phenotypes predicted by its genome-scale parent. These unrealistic predictions from *i*ML1515 are not the result of "errors" in the metabolic model. Rather, they are the result of applying simple stoichiometric methods, such as FBA, to a large network with many degrees of freedom. It is possible that the inclusion of additional constraints, such as thermodynamic feasibility under physiological conditions or proteome allocation bounds, would automatically render such solutions infeasible. However, these constraints, and the parameters required to implement them, are not always readily available. By assembling a smaller model and curating it with expert knowledge, we filter out many of these behaviours by construction, providing users with a versatile and interpretable tool to investigate central and biosynthetic metabolism in *E. coli*. Clearly, this comes at the cost of limited applicability in other scenarios, such as those where the metabolic subsystems not included in *i*CH360 (e.g. degradation pathways) are crucial to explaining or modelling a given phenotype. In these cases, the use of a genome-scale model (or an *ad hoc* reduction thereof) would still be an invaluable tool.

To make this model easily usable in a variety of applications, we enriched the stoichiometric network structure with a curated layer of biological knowledge in the form of a knowledge graph. This graph encodes information about biological entities in the network in a structured, ready-to-use format, including catalytic relationships between reactions and enzymes, the stoichiometric composition of protein complexes, and small-molecule regulation interactions. In addition, we mapped to the model a range of quantitative parameters, including *in vivo* turnover number estimates and thermodynamic constants, extending the use of the model beyond a simple stoichiometric analysis. A summary of the biological knowledge captured by *i*CH360 is shown in Table 3.

Due to its medium size and the high level of curation, *i*CH360 lends itself to a wide range of modelling methods. Here, we demonstrated some representative examples. These include the calculation of production envelopes, the modelling of metabolic proteome allocation via the enzyme-constraint model variant EC-*i*CH360, enumerating and analysing elementary flux modes in the network via the reduced model variant *i*CH360$_{red}$, and performing thermodynamic-based analysis using the set of thermodynamic constants provided. Nevertheless, we believe that many other analyses are possible. For example, alternative definitions of elementary pathways, such as elementary conversion modes, would be valuable to explore as a more tractable alternative to elementary flux modes [39]. Similarly, our analyses of metabolic enzyme cost presented here have relied on simple, capacity-based definitions of enzyme cost, where a constant cost per unit flux is assumed for each reaction regardless of the condition studied. However, with additional kinetic parametrisation of the model, more complete enzyme cost functions, explicitly accounting for condition-specific metabolite concentrations and, consequently, enzyme costs, could be used to generate more realistic estimates of metabolic tradeoffs predicted by the model [32].

Indeed, while at this stage the parametrisation of the model is limited to turnover numbers and thermodynamic constants, we anticipate that additional parameter sets can easily be mapped to the model. Facilitated by extensive annotations present in the model and by the recent development of machine learning-enabled kinetic constant estimators [40–43], a complete kinetic parametrisation of *i*CH360 is thus a valuable potential future development. In addition, probabilistic estimates of the kinetic parameters [43,44] can be combined with our existing thermodynamic parametrisation, making it possible to account for (potentially correlated) parameter uncertainty throughout the kinetic modelling process.

**Table 3.** **A summary of knowledge captured by the *i*CH360 model, as well as example simulations and analyses shown in this article.**

| Model structure | Notes | Data source |
|---|---|---|
| Network (reaction stoichiometries) | Selected reactions from *i*ML1515, hand-curated | [8] |
| Annotations to external databases | Parsed from *i*ML1515, extended, and updated | [8], manual curation |
| Network graphics | Escher maps of the full model and its subsystems | |
| Biological knowledge supporting the stoichiometric model | Catalytic relationships, protein complex composition, small-molecule regulations, and others | [24] |
| **Physico-chemical parameters mapped to model** | | |
| Thermodynamic constants ($\Delta_r G'^\circ$) | Account for compartment-specific chemical environment (pH, pMg, ionic strength, and potential) and include corrections for reactions occurring across compartments. | [35], manual curation |
| $k_{\mathrm{app}}^{\max}$ values | *in vivo* estimates of catalytic turnover numbers in units of s$^{-1}$ | [30] |
| Typical $k_{\mathrm{app}}$ values | Adjusted estimates of turnover numbers (in units of s$^{-1}$), fitted to proteomic data, accounting for typical saturation levels across growth conditions | [30] [31] |
| Protein molecular masses | Molecular masses for all proteins/protein complexes covered by the model | [24] |
| **Cell-state data mapped to the model** | | |
| Protein abundances | Abundance across different growth conditions for proteins/protein complexes covered by the model, estimated from experimentally measured polypeptide abundances | [31] |
| Metabolite concentrations | Measured metabolite concentrations across growth conditions | [45] |
| Metabolic fluxes | Measured metabolite fluxes for aerobic growth on glucose | [30], [46] |
| **Example applications shown** | | |
| Production envelope analysis | See Fig 2 and Fig H in S1 Text | |
| Enzyme-constrained FBA | With enzyme-constrained version of the model, EC-*i*CH360, constructed in sMOMENT format [28] | |
| EFM analysis Saturation FBA | With reduced model variant *i*CH360$_{\mathrm{red}}$. Predicts metabolic fluxes as a function of external substrate concentration | |
| Max-Min Driving force | Formulation from [36], extended to account for correlated uncertainty in the thermodynamic constants estimates. | |
| Probabilistic Metabolic Optimisation | Prediction of thermodynamic states (metabolite concentrations and relative fluxes) maximally consistent with measured metabolite concentrations [37] | |

In light of the above results, we believe that *i*CH360 has the potential to become a reference metabolic model for *E. coli*.

## Materials and methods

### Model assembly and curation

All relevant pathways included in the model (the core metabolism reactions from ECC [15] and the biosynthesis pathways shown in Figs A-D in S1 Text) were assembled and curated based on information available in the EcoCyc [24] and KEGG [22] databases. The respective reactions were then extracted from *i*ML1515 and parsed into a new model. To compute an equivalent biomass reaction, we first collected all the pathways required for the production of the components present in the *i*ML1515 biomass reaction (with the exception of a small number of compounds present with very small stoichiometry, which were neglected from this analysis for simplicity), but not in our model. We used the "core" biomass reaction from *i*ML1515 (BIOMASS_Ec_iML1515_core_75p37M), rather than alternative "WT" reaction (BIOMASS_Ec_iML1515_WT_75p37M), since its smaller number of requirements made it easier to manually curate the pathways for their production. These additional pathways (available in the repository supporting this manuscript) were manually curated based on the available literature and database annotations to ensure they represent the biologically most relevant bioproduction route for each biomass component. By adding these pathways to *i*CH360, we obtained an extended model that was able to predict growth rates directly

through the original *i*ML1515 biomass reaction. The equivalent biomass function was then calculated based on a reference flux distribution computed on this extended model, as explained in Sect B of S1 Text. Both growth-associated (the stoichiometry of intracellular ATP in the biomass reaction) and non-growth-associated (the lower bound on the maintenance reaction, ATPM) energy requirements were directly inherited from *i*ML1515. Model assembly, manipulation and validation were performed using COBRApy[18]. The extension of database cross-annotation for the model reactions was performed through a mixture of automated database query and manual curation.

## Network graphics

*i*CH360 can be visualised through a series of custom-built maps using the metabolic visualisation tool Escher [19] (see Fig 1). There are three main ways to visualise the model or solutions thereof. First, a complete map of the model, including all of its reactions, can be used (Fig 1). In order to provide a more compact representation of the network, a compressed second variant of the same map was constructed (Fig U in S1 Text). Here, long biosynthetic linear pathways were lumped into single pseudo-reactions, which only show the net production or consumption of metabolites by the pathway while omitting intermediates. Finally, individual maps are provided for each of the main subsystems in the model.

## Production envelope analysis

All production envelopes shown in the main text and Supporting Information were generated using the built-in production envelopes tools from the `cobra.flux_analysis. phenotype_phase_plane` module in the COBRApy package [18]. Briefly, the algorithm first computes the maximum and minimum production rates of the metabolite of interest, given the existing constraints in the model (including the specified bounds on the uptake of the carbon source and oxygen). The interval between the maximum and minimum achievable production rates is then discretised into an equally spaced grid of points. For each point in the interval, the production rate is fixed and the model's objective (here, the growth rate) is sequentially maximised and minimised, thus generating the boundary of the production envelope. All production envelopes were computed by specifying an upper bound on the uptake of the carbon source of 10 mmol/gDW/h, and blocking oxygen uptake for the anaerobic scenario. For comparisons with ECC and ECC2, the maintenance requirement (lower bound on the ATPM reaction) of these two models was set to the same value used in *i*ML1515 and *i*CH360 (6.86 mmol/gDW/h).

## Knowledge graph for linking reactions to enzymes and proteins

Information about the enzymes and proteins behind the stoichiometric model was collected in a knowledge graph. To build this graph, all available data on reaction-protein association and subunit composition were retrieved by automatically querying the BioCyc database through its REST-based data retrieval API (https://biocyc.org/web-services.shtml). This information was then extended and curated based on a comparison with existing *i*ML1515 GPR annotations. The resulting data were used to generate a directed graph in which nodes represent biological entities (such as reactions, proteins, genes, and metabolites) and edges represent functional dependencies across them, including catalysis, subunit composition, post-translational modifications, and others. A complete list of node and edge types in the graph is provided, respectively, in Tables B and C in S1 Text. All polypeptide nodes were annotated with their molecular mass (parsed from EcoCyc), enabling recursive computation

of the molecular mass of any protein node in the graph (see Sect D in S1 Text). All manipulation and analysis of the graph data structure were performed using the NetworkX Python package [47], and the final data structure is provided to the user in Cytoscape and GML formats.

## Primary and secondary catalytic edges

Catalytic edges in the graph, i.e. edges connecting reaction and enzyme nodes, were manually annotated as either primary or secondary, based on available evidence for the activity of each enzyme in the model with respect to its associated reactions (File B in S1 Data). More specifically, a catalytic edge between a reaction-enzyme pair was labelled as secondary whenever the enzyme was shown in the literature to account for only minor catalytic activity for a reaction when compared with another isozyme. In this case, references to the relevant literature were included as metadata for that edge. Whenever sufficient information was not available, all isozyme edges for a reaction were conservatively treated as primary.

## Catalytic disruption analysis

For the catalytic disruption test, we considered a total of 9 growth conditions (aerobic growth on glucose, fructose, lactate, pyruvate, succinate, acetate, glycerol, ribose, and xylose) and identified condition-specific essential reactions in the following way: in the model, for each condition considered, we determined the reactions whose knockout led to the inability to produce biomass. Two anecdotally known false-essential reactions (the glycine cleavage system, GLYCL, and thioredoxin reductase, TRDR), which are essential in *i*CH360 only due to its lack of certain reactions or pathways, were excluded from the analysis. GLYCL is essential in *i*CH360 since it lacks a THF-charging reaction, FTHFLi, present in the genome-scale parent. TRDR is essential in our model because a number of reactions using glutaredoxin as a cofactor are not included, making thioredoxin regeneration essential in our model, but not in practice. For each essential reaction, we enumerated the associated genes and simulated their individual knockouts. Knockouts were propagated across the knowledge graph using Boolean logic rules (see Sect D in S1 Text), deactivating proteins and reactions as appropriate. Each resulting gene-condition pair was labelled with a disruption class depending on the type of enzyme-reaction associations lost: complete, full primary, partial primary, or secondary (see Results). If a gene knockout affected multiple reactions, we labelled the gene with the most severe disruption across all affected reactions in the following precedence order: complete disruption, full primary disruption, partial primary disruption, secondary disruption. Finally, we compared predicted disruption types to measured fitness values from Price et al. (2018) [27], using Wilcoxon rank-sum tests to assess statistically significant differences in fitness between disruption classes.

## Construction of the enzyme-constrained metabolic model

In the sMOMENT formalism, the positive and negative fluxes in each reaction are formally described, respectively, as positive fluxes in separate "forward" and "backward" versions of the reaction. To construct the enzyme-constrained model in the sMOMENT format, reversible reactions in the model were duplicated to separately represent fluxes in the forward or backward direction. Direction-specific turnover number estimates were parsed from Heckmann et al. (2020) [30] and a default value of 65 s$^{-1}$ was used for transporters, as in [48]. To account for the fact that [30] reports values as turnover numbers per polypeptide, the values were

multiplied by the number of polypeptide subunits in each enzyme. For each reaction-enzyme pair, an enzyme cost per unit flux (in units of $g \cdot h \cdot mmol^{-1}$) was then defined as:

$$a_i = \frac{M_i}{k_{\text{cat},i}\, \sigma} \tag{1}$$

where $a_i$ is the enzyme cost of reaction-enzyme pair $i$, $k_{\text{cat},i}$ is the turnover rate estimate for the pair (here, in units of $h^{-1}$), $M_i$ is the molecular mass of the enzyme involved (in kDa), and $\sigma$ is a unitless condition-specific scaling factor (typically interpreted as an average enzyme saturation value). Then, a unique enzyme was assigned to each reaction. To this end, secondary catalytic relationships were first discarded; for reactions with multiple annotated primary isozymes, the enzyme with the highest measured abundance in the integrated PAX Database [49] was heuristically chosen. Based on these costs per unit flux, an enzyme capacity constraint was introduced as:

$$\sum_i a_i\, v_i \leq e_{\text{tot}} \tag{2}$$

where $v_i$ denotes the flux in reaction $i$ (in mmol/gDW/h) and $e_{\text{tot}}$ is a parameter denoting the total amount of enzyme mass (in g/gDW) that can be allocated to the flux mode. The constraint is enforced by augmenting the stoichiometric matrix of the model with an additional enzyme supply pseudoreaction (upper-bounded by $e_{\text{tot}}$) and an enzyme pool pseudometabolite consumed in each reaction with stoichiometry $a_i$ [28].

## Adjustment of turnover numbers across conditions

Turnover numbers from Heckmann et al. (2020) [30] were adjusted based on the nonlinear programming (NLP) formulation detailed in Sect F of S1 Text. The reference flux distributions required by the procedure were obtained using the original (unadjusted) set of turnover numbers and the bounds on allowable adjustments ($\mathbf{u}_{\min}$ and $\mathbf{u}_{\max}$ in Eq (19) of S1 Text) set to $\pm 2$ (corresponding to a maximal 100-fold increase or reduction of each parameter from the original value). The linear program used to obtain reference flux distributions was formulated and solved with GUROBI [50], while the nonlinear program used for turnover adjustment was formulated and solved with the open-source optimisation package CasADi [51]. To investigate the effect of the ridge regularisation hyperparameter ($\rho$ in in Eq (19) of S1 Text), we solved the adjustment problem for a broad range of values of this parameter and, each time, we computed the RMSE between measurements and predictions of enzyme abundance, as well as the mean absolute deviation between original and adjusted turnover values, both computed for $\log_{10}$-transformed data (Fig M in S1 Text). Based on this information, a value of $\rho = 1$, after which any further decrease in the amount of regularisation results in marginal reduction of the RMSE, was heuristically chosen to compute the final set of adjusted turnover numbers. Upon reparametrisation into apparent turnover numbers (see Sect F in S1 Text), this set of adjusted parameters was used to parametrise the enzyme-constrained model variant EC-*i*CH360. For the leave-one-out cross validation, the adjustment procedure was run multiple times, each time excluding abundance data from a condition from the training dataset. The resulting adjusted parameter set was then used to generate enzyme abundance predictions for the condition left out, which were then compared against measured values.

## Enzyme allocation predictions

To validate enzyme allocation predictions against experimental values, we first retrieved measured polypeptide abundances for each growth condition from [31] and imputed missing values using, whenever available, abundance values from the PAX Database [49]. Then we estimated enzyme counts across conditions from polypeptide counts using non-negative least-squares estimation (Sect E in S1 Text). Next, we converted them into mass abundances (in units of g/gDW) based on the molecular mass of each complex (see Sect D in S1 Text) and assuming a cell dry mass of $2.8 \times 10^{-13}$g (BIONUMBER ID 103904 [52]). Enzyme allocation predictions for each condition were then computed via EC-*i*CH360 by fixing the growth rate to the experimentally measured one from [31] and minimising the total enzyme cost, initially using a value of 1 for the average saturation parameter $\sigma$ (Eq (1)). For the predictions computed using the turnover number estimates from Heckmann et al. (2020) [30], the average saturation coefficient $\sigma$ was then estimated from data, for each condition, as:

$$\sigma = \frac{\sum_{i \in \mathcal{M}} e_i}{\bar{e}_{\text{tot}}} \tag{3}$$

where $\mathcal{M}$ denotes the index set of model enzymes for which measurements are available, $e_i$ is the predicted abundance for the $i$th enzyme, and $\bar{e}_{\text{tot}}$ is the total measured model enzyme abundance for that condition. This value of $\sigma$ was then used to scale the predicted enzyme abundances before comparing them with the ones measured experimentally. Note that this choice of $\sigma$ ensures that the sum of predicted abundances matches that of measured ones. To compute predictions with the adjusted parameter set, the scaling factors for each condition were obtained as part of the fitting procedure (see Sect F in S1 Text). All reported root mean squared errors were computed on $\log_{10}$ transformed enzyme abundances, excluding enzymes with zero predicted abundance from the computation.

## Enumeration of elementary flux modes

Elementary Flux Modes (EFMs) for the submodel $i$CH360$_{\text{red}}$ were enumerated, for each growth condition, using EFMtools [53]. Filtered modes (Table E in S1 Text) were defined as those supporting non-zero biomass flux and, for aerobic modes, non-zero oxygen uptake. In addition, in the aerobic case, filtered modes exclude those carrying flux in either of three reactions – Pyruvate-Formate Lyase (PFL), Fumarate Reductase (FRD2), and the menaquinone-dependent Dihydroorotate Dehydrogenase (DHORD5) – known to be physiologically active only under anaerobic conditions [54–56].

## Growth/yield trade-off analysis

To analyse trade-offs between growth rate and yield, we computed the yield and a predicted growth rate of each EFM, using the unit conventions common in stoichiometric metabolic models. The specific growth rate $\mu$ (in h$^{-1}$) of a cell or a cell population can be defined as the rate of biomass production per unit of biomass present. In stoichiometric metabolic models, units are chosen in a specific way such that the growth rate is identically given by the rate $v_{\text{BM}}$ of the biomass reaction. In FBA models, normal reaction fluxes are given in units of mmol/gDW/h, so the yield of an EFM, computed as the ratio of its biomass flux $v_{\text{BM}}$ to its glucose uptake flux, has a unit of gDW/mmol. The biomass flux, whose conventional unit is h$^{-1}$ was directly interpreted as the cell growth rate $\mu$. To determine the cell growth rate allowed by a flux distribution, we first computed its absolute enzyme cost

$$c_{\text{enz}} = \sum_i a_i \, v_i \tag{4}$$

where $c_{\text{enz}}$ is the enzyme cost of the flux distribution (an enzyme mass, measured in g/gDW), $v_i$ is the flux of reaction $i$, and $a_i$ is the enzyme cost per unit flux in reaction $i$, computed as per Eq (1) using the set of adjusted turnover numbers. Since an elementary flux mode can be scaled arbitrarily, both $v_{\text{BM}}$ and $c_{\text{enz}}$ depend on the particular choice of scaling of the mode (though their ratio, $v_{\text{BM}}/c_{\text{enz}}$, does not). In order to obtain an estimate for the achievable growth rate of a mode $\mu$ that is a unique property of each EFM, independent of its scaling, we thus normalised all modes to the same total enzyme cost $f_{\text{enz}}$ (in g/gDW), and looked at the resulting flux through the biomass reaction. More formally, the achievable growth rate $\mu$ (in h$^{-1}$) for an (arbitrarily scaled) EFM with biomass flux $v_{\text{BM}}$ and enzyme cost $c_{\text{enz}}$ was computed as:

$$\mu = f_{\text{enz}} \, \frac{v_{\text{BM}}}{c_{\text{enz}}} \tag{5}$$

where $f_{\text{enz}}$ denotes the total mass of enzyme available for the flux distribution, relative to the total dry mass of the cell. For simplicity, we approximate $f_{\text{enz}}$ by a constant value of $f_{\text{enz}} = 0.285$ g/gDW, which we obtained by taking the minimum enzyme investment required by the enzyme-constrained model to support the experimentally measured growth rate reported in the proteomic dataset [31] for the condition of interest in this analysis (aerobic growth on glucose).

To simulate low-oxygen conditions, the cost per unit flux of all oxygen-consuming reactions (CYTBO3_4pp, CYTBDpp, CYTBD2pp) was increased by a 1000-fold, mimicking the physiological state in which these reactions operate at low saturation with oxygen.

## Saturation FBA analysis

Saturation FBA calculations were performed by optimising biomass production in the enzyme-constrained model, setting the saturation coefficient ($\sigma$ in Eq (1)) to the value fitted as part of the turnover number adjusting procedure (Methods) for all reactions except the glucose transporter (GLCptspp). The saturation of the glucose transporter, $\sigma_{\text{up}}$, was computed as a function of external glucose concentration and assuming irreversible Michaelis-Menten kinetics, so that:

$$\sigma_{\text{up}} = \frac{[\text{Glc}]}{K_{\text{m}} + [\text{Glc}]} \tag{6}$$

where $[\text{Glc}]$ is the external glucose concentration (in mM) and a value of 0.116 mM was used for the Michaelis constant $K_{\text{m}}$ [13].

## Component contribution estimates of thermodynamic constants

Estimates of the free energies of reactions, and their uncertainties, were obtained using the component contribution framework previously described in [35]. Several reactions in the model involve protein side groups, such as the acyl-carrying protein (ACP), or cofactors, such as glutaredoxin, for which a decomposition in terms of chemical groups is not available. As a result, thermodynamic constants for these reactions cannot be directly estimated through database-based implementations of the component contribution method, such as eQuilibrator [57]. However, since these non-decomposable protein groups are conserved in all reactions within our model, their net contribution to the reaction thermodynamics is, at least from a group contribution perspective, null. If we were only interested in computing standard

free energies of reaction, $\Delta_r G°$, we could simply treat protein groups as non-decomposable "black-box" units and add them to the group incidence matrix of the eQuilibrator database (see Sect S1.1 in [57]), enabling us to construct a group decomposition for compounds that contain them. However, for the computation of transformed standard free energies of reaction, $\Delta_r G'°$, an exact chemical definition of each metabolite is required.

To address this issue, protein groups were replaced by an appropriate chemical moiety that best approximates the metabolite's chemical environment. Specifically, ACP groups were replaced by a phosphopantetheine group, the natural prosthetic group of acyl-carrier proteins, with a methyl group at the attachment site to the protein scaffold. Similarly, glutaredoxin was replaced by its Cys-Pro-Tyr-Cys active site, with the two cysteines being either free (for the reduced form of the cofactor) or linked by a disulfide bridge (for the oxidised state). International Chemical Identifiers (InChI) were constructed for these "replacement" metabolites (File C in S1 Data) and used to extend the default compound cache of the eQuilibrator database. This custom-extended eQuilibrator compound cache is available in the repository supporting this manuscript.

Corrections for reactions that occur in different compartments were calculated as described in [57], using compartment-specific pH, pMg, ionic strength, and potentials from [43].

## Max-min driving force computation

The max-min driving force (MDF) of each reference flux distribution was calculated by extending the original formulation described in [36] to account for correlated uncertainty in the estimates of the thermodynamic constants. Let $\Delta_r G'° \in \mathbb{R}^N$ be a random vector representing the (uncertain) standard Gibbs free energy of reaction for the reactions in the network. Importantly, this vector includes only balanced metabolic reactions and excludes pseudoreactions such as exchange reactions. To describe our uncertain knowledge, we assume that the vector $\Delta_r G'°$ follows a multivariate normal distribution:

$$\Delta_r G'° \sim \mathcal{N}(\overline{\Delta_r G'°}, \Sigma) \tag{7}$$

where $\overline{\Delta_r G'°}$ and $\Sigma$ are the mean vector and covariance matrix of the estimates obtained through the component contribution methods. This random vector can equivalently be expressed as:

$$\Delta_r G'° = \overline{\Delta_r G'°} + \mathbf{Q}\,\mathbf{z} \tag{8}$$

where $\mathbf{z}$ is a standard normal random vector in $\mathbb{R}^q$, with $q = \text{rank}(\Sigma)$, and $\mathbf{Q} \in \mathbb{R}^{N \times q}$ is a square root of the covariance matrix, i.e. it satisfies:

$$\Sigma = \mathbf{Q}\,\mathbf{Q}^T. \tag{9}$$

In order to integrate this probabilistic description within a typical constrained-optimisation formulation, we define the set $\mathcal{D}_\alpha$ as the $\alpha$-level confidence region around the mean of $\Delta_r G'°$. Using Eq (7), and noting that the squared norm of a standard normal random variable is known to be Chi-squared distributed, we can represent this set as:

$$\mathcal{D}_\alpha = \left\{ \mathbf{x} \in \mathbb{R}^N \mid \mathbf{x} = \overline{\Delta_r G'°} + \mathbf{Q}\,\mathbf{m}, \quad ||\mathbf{m}||_2^2 \le \chi_{q;\alpha}^2 \right\} \tag{10}$$

where $\mathbf{m} \in \mathbb{R}^q$ is a vector of free parameters and $\chi_q(.)$ denotes the quantile function (inverse cumulative distribution function) of a chi-squared distribution with $q$ degrees of freedom. We can thus account for the uncertainty in thermodynamic estimates by treating the free energies of reaction as decision variables, rather than known parameters, and constraining their value to belong to $\mathcal{D}_\alpha$. For a given reference flux distribution $\mathbf{v}$, this results in the following quadratically-constrained program (QCP):

$$
\begin{aligned}
\max_{\mathbf{m},\mathbf{c},b} \quad & b \\
\text{s.t.} \quad & \Delta_r G'^\circ = \overline{\Delta_r G'^\circ} + \mathbf{Q}\,\mathbf{m} \\
& \Delta_r G' = \Delta_r G'^\circ + \mathrm{RT}\,\mathbf{S}^\top \mathbf{c} \\
& -\operatorname{sign}(v_i)\,\Delta_r G_i' > b, \qquad \text{if } v_i \neq 0 \\
& \|\mathbf{m}\|_2^2 \le \chi_{q;\alpha}^2 \\
& \mathbf{c}_{\min} \le \mathbf{c} \le \mathbf{c}_{\max}
\end{aligned}
\tag{11}
$$

where $b$ is the min driving force (or the MDF after optimisation), $\mathbf{c} \in \mathbb{R}^m$ is a vector of log-metabolite concentrations, $\mathbf{c}_{\min}; \mathbf{c}_{\max} \in \mathbb{R}^m$ are lower and upper bounds on these log-concentrations, $\mathbf{S} \in \mathbb{R}^{m \times N}$ is the stoichiometric matrix of the model, R is the ideal gas constant, and T is the temperature used for the computation of the free energy estimates. For our MDF calculations, we used a confidence level of 90% and set the bounds on the concentration of all metabolites to the physiologically plausible range of (1 $\mu$M, 10 mM). The above quadratically constrained program was formulated and solved using the GUROBI package [50].

## Probabilistic thermodynamic analysis

Probabilistic metabolic optimisation (PMO) of the model was performed using the PTA Python package [37], providing the software with the curated thermodynamic estimates generated for this model. The default values available through the package for growth in M9 medium with glucose (which include measurements from [45]) were used as priors for the concentration of metabolites, and the growth rate was bounded from below by the reported value in [45]. Furthermore, for this analysis, which requires that each flux in the solution have a well-defined directionality, two transhydrogenase reactions (NADH17pp and THD2pp) were allowed to operate reversibly [37].

To analyse the thermodynamic state computed by PTA, we compute the flux-force efficacy $\eta$ for each reaction as (see Fig 8):

$$
\eta = \frac{e^{-\frac{\Delta_r G'}{\mathrm{RT}}} - 1}{e^{-\frac{\Delta_r G'}{\mathrm{RT}}} + 1} = \tanh\left(-\frac{1}{2}\frac{\Delta_r G'}{\mathrm{RT}}\right)
\tag{12}
$$

where $\Delta_r G'$ is the Gibbs free energy of reaction in the thermodynamic state, R is the universal gas constant, and T is the temperature considered for the analysis (310.15 K). To compare the PTA-predicted flux force efficacies with experimental measurements of enzyme abundance, we selected reactions that carried at least 2.5% of the glucose uptake flux and pooled them into two groups, corresponding to $\eta > 0.5$ (72 reactions) and $\eta < 0.5$ (31 reactions). $\eta = 0.5$ corresponds to $\Delta_r G' \approx -1.1\,\mathrm{RT} = -2.8\,\mathrm{kJ/mol}$.

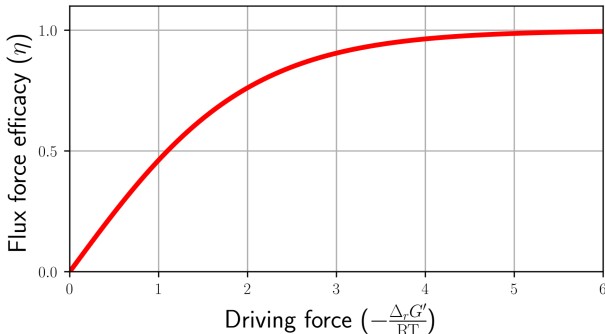

**Fig 8. The flux force efficacy ($\eta$) as a function of the (scaled) negative Gibbs free energy of reaction, $-\frac{\Delta_r G'}{RT}$.** The efficacy of the flux force corresponds to the ratio between the net flux (forward minus backward flux) and the total flux (forward plus backward flux) of a reaction, which approaches 1 for reactions operating far from chemical equilibrium ($\Delta_r G' \ll 0$).

## Supporting information

**S1 Text. Supplementary material text, including Sects A–F, Figs A–V, and Tables A–E.**
(PDF)

**S1 Data. Supplementary Files A–C .**
(XLSX)

## Author contributions

**Conceptualization:** Arren Bar-Even.

**Formal analysis:** Marco Corrao, Hai He, Wolfram Liebermeister, Elad Noor.

**Investigation:** Marco Corrao.

**Methodology:** Marco Corrao, Hai He.

**Software:** Marco Corrao.

**Supervision:** Hai He, Wolfram Liebermeister, Elad Noor, Arren Bar-Even.

**Validation:** Marco Corrao.

**Visualization:** Marco Corrao.

**Writing – original draft:** Marco Corrao, Hai He, Wolfram Liebermeister, Elad Noor.

**Writing – review & editing:** Marco Corrao, Hai He, Wolfram Liebermeister, Elad Noor.

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
