## [Decision Letter · Decision Letter 0]

12 Aug 2025

PCOMPBIOL-D-25-01048

A compact model of *Escherichia coli* core and biosynthetic metabolism

PLOS Computational Biology

Dear Dr. He,

Thank you for submitting your manuscript to PLOS Computational Biology. After careful consideration, we feel that it has merit but does not fully meet PLOS Computational Biology's publication criteria as it currently stands. Therefore, we invite you to submit a revised version of the manuscript that addresses the points raised during the review process.

Please submit your revised manuscript within 60 days Oct 12 2025 11:59PM. If you will need more time than this to complete your revisions, please reply to this message or contact the journal office at ploscompbiol@plos.org. Please include the following items when submitting your revised manuscript:

We look forward to receiving your revised manuscript.

Kind regards,

Claudio Angione

Academic Editor

PLOS Computational Biology

Marc Birtwistle

Section Editor

PLOS Computational Biology

**Additional Editor Comments:**

The reviewers find your paper of interest, but they also raise some important points that should be addressed carefully in a revised version.

**Journal Requirements:**

At this stage, the following Authors/Authors require contributions: Marco Corrao, Hai He, Wolfram Liebermeister, Elad Noor, and Arren Bar-Even. Please ensure that the full contributions of each author are acknowledged in the "Add/Edit/Remove Authors" section of our submission form.

5) We have noticed that you referred to Supplementary Files (1-3) in the manuscript. However, there are no corresponding files uploaded to the submission. Please upload them as separate files with the item type 'Supporting Information'.

6) We have noticed that you have uploaded Supporting Information files, but you have not included a list of legends. Please add a full list of legends for your Supporting Information files after the references list.

7) Your current Financial Disclosure states, "The author(s) received no specific funding for this work.".

However, your funding information on the submission form indicates receiving a fund. Please ensure that the funders and grant numbers match between the Financial Disclosure field and the Funding Information tab in your submission form. Note that the funders must be provided in the same order in both places as well. 

Please amend your detailed Financial Disclosure statement. This is published with the article. It must therefore be completed in full sentences and contain the exact wording you wish to be published.

1) Please clarify all sources of financial support for your study. List the grants, grant numbers, and organizations that funded your study, including funding received from your institution. Please note that suppliers of material support, including research materials, should be recognized in the Acknowledgements section rather than in the Financial Disclosure

2) State the initials, alongside each funding source, of each author to receive each grant. For example: "This work was supported by the National Institutes of Health (####### to AM; ###### to CJ) and the National Science Foundation (###### to AM)."

3) State what role the funders took in the study. If the funders had no role in your study, please state: "The funders had no role in study design, data collection and analysis, decision to publish, or preparation of the manuscript."

4) If any authors received a salary from any of your funders, please state which authors and which funders.

8) Currently, your Competing Interest statement on the submission form is "The authors declare no financial conflicts of interest." please modify this to the standard "The authors have declared that no competing interests exist" at resubmission.

**Comments to the Authors:**

**Please note that one of the reviews is uploaded as an attachment.**

**Reviewers' comments:**

Reviewer's Responses to Questions

Reviewer #1: Genome-scale models of microbial metabolism offer a wide range of applicability from understanding to engineering the metabolism. However, the large size of these models can hinder its interpretability and visualization. To address these issues, authors had developed a compact model of Escherichia coli K-12 MG1655 (iCH360) extracted from the existing reconstruction, iML1515. One of the key contributions of the paper include enriching the network with a weighted knowledge graph constituting the reactions, proteins, genes and compounds as nodes and annotation of the catalytic relationship as either primary or secondary that together improved the gene essentiality predictions. Furthermore, development the escher maps can help the modellers visualize the flux distribution across a wide range of environmental conditions. Extensive computational experiments done on iCH360, shows its alignment and distinction compared to its parent model, iML1515. Overall, the compact model of E.coli is a valuable tool to the modelling community. However, this manuscript itself might benefit from a few modifications suggested below.

Major comments:

1) Can the authors explain the reason for choosing Ethanol, Acetate, Lactate and Succinate in the production envelop analysis? How will the results change if any other products are chosen for this analysis?

2) In computing equivalent biomass reaction, does the coefficient of the biomass reaction vary based on alternate solutions that can be obtained in FBA (Multiple solutions possible for v*). Authors have mentioned that the biomass reaction will remain unique across the conditions. Can they shed more light on this? Has not even a single coefficient has changed across the conditions? If changed can the authors quantify it across the conditions and alternate solutions possible in FBA.

3) At line-507 the authors mention “A small number of known false positives, which are essential in iCH360 only due to its lack of certain reactions or pathways, were excluded from the analysis.” Can the authors explain how they got these false positives? In addition, list them out?

4) Authors have used pFBA solutions to calculate the MDF and report that all the flux distribution have positive MDF highlighting their thermodynamic feasibility. However using pFBA will always remove loops in the flux distribution (Refer: https://academic.oup.com/bioinformatics/article/31/13/2159/195895) given that the objective function does not participate in any infeasible cycles. Can the authors shed some light on how the MDF value change on using FBA? Also, is it even possible to have a negative value for MDF on using pFBA?

5) Escher maps are already available for e.coli metabolism. What is the key contribution of the new maps build here?

Minor comments:

1) There are lot of spelling mistakes in the paper. Few examples: Line 172: estensive; Line 189: catalyses; Supplemental section A3: man text

2) Line: 226. Which section are the authors referring to?

3) The authors use COBRApy not COBRA toolbox. Modify it in lines 82 and 459.

4) Italics was used very often in unnecessary places

5) Most of the core (or compact) models developed are for prokaryotic models especially e.coli models. Can the authors discuss how these methods can be extended to apply on eukaryotic models?

Reviewer #2: This is an elegant and well-written paper that presents an expertly curated, medium-scale metabolic model of Escherichia coli. The quality of the work is impressive, and the resulting model will undoubtedly support numerous future applications. My comments have been uploaded as an attachment.

Reviewer #3: Peer Review Report for Manuscript PCOMPBIOL 25-01048

Recommendation: Accept with Minor Revisions

General Comments:

This manuscript presents a nicely developed, annotated stoichiometric model for E. coli, enriched with well-motivated additions such as thermodynamics and resource allocation modeling. It represents a valuable contribution to the metabolic engineering community and broadens the toolkit available for metabolic modeling studies.

Specific Comments and Suggestions:

Abstract:

- L9: The text mentions E. coli K-12 MG1655. Since the model is reduced, its applicability likely extends beyond this strain. Consider emphasizing the broader applicability and potential for adaptation to other model organisms with suitable modifications.

- Query: Address if the model simulations replicate the growth defects observed in the study by [Reference: 10.1016/j.cels.2020.10.011] with CRISPRi targeting CysH, MetE, Ppc, and Pts.

- L18: While comparisons to genome-scale models are mentioned, what are the advantages over the core (ECC/ECC2) model.

Minor comments:

- L50: Substantiate the claim regarding "difficulties with constraints in genome-scale" models by citing relevant literature, such as:

- [Reference: https://doi.org/10.1016/j.copbio.2015.08.021]

- [Reference: https://doi.org/10.1016/j.mib.2010.03.001]

- Include references for refined genome-scale models:

- [Reference: 10.1529/biophysj.106.093138]

- [References: https://doi.org/10.1007/978-1-4939-1170-7_3], etc.

- L71: Clarify the term "effective" in the context of biomass-producing reactions; consider replacing it with "compact" or "lumped" for precision.

- L87: Explain why not all five nucleotides were included in the model.

- L126-137 and L141: Consider moving detailed analysis and figures to supplementary materials to maintain the flow of the main text. Confirm if thermodynamic statements were verified by calculations.

- Fig. 2: Comment on the high growth rate of 0.9 in minimal medium. Consider expressing the x-axis in yield terms for better comparison with experimental data.

- Fig. 3D: Define the black dots and clarify fitness definition. Was it growth rate in relation to WT growth?

- "Primary and secondary catalysis": Specify conditions, and clarify if there was a quantitative cut-off.

- L230: Explain the treatment of lumped reactions concerning turnover numbers in both the model and experimentally.

- L241: Describe the "custom heuristic" method or approach.

- Fig. 4: Elaborate on how lumped pathways are represented and discuss observed shifts in enzyme abundance predictions vis-à-vis experimental data.

- Fig. 5 & 6: Address discrepancies in growth rates observed across figures, clarify assumptions on maintenance, and suggest consistent units (e.g., g/gGlc) for ease of comparison.

- L363: Clarify how substrate channeling influences thermodynamics. Discuss potential reaction coupling mechanisms.

- Table 1: Explain the exclusion of arabinose, despite its significance in synthetic biology.

Technical:

- Ensure the abbreviation list is comprehensive, particularly for enzymes.

**Have the authors made all data and (if applicable) computational code underlying the findings in their manuscript fully available?**

Reviewer #1: Yes

Reviewer #2: Yes

Reviewer #3: None

PLOS authors have the option to publish the peer review history of their article (what does this mean?). If published, this will include your full peer review and any attached files.

Reviewer #1: **Yes: **Pavan Kumar S

Reviewer #2: **Yes: **Ghjuvan Grimaud

Reviewer #3: No

**Figure resubmission:**
---

## [Decision Letter · Decision Letter 1]

28 Sep 2025

Dear Dr. He,

We are pleased to inform you that your manuscript 'A compact model of *Escherichia coli* core and biosynthetic metabolism' has been provisionally accepted for publication in PLOS Computational Biology.

Best regards,

Claudio Angione

Academic Editor

PLOS Computational Biology

Marc Birtwistle

Section Editor

PLOS Computational Biology

Reviewer #1:

Reviewer #2:

Reviewer #3:

Reviewer's Responses to Questions

**Comments to the Authors:**

Reviewer #1: I am satisfied with the efforts of the authors and agree to publish the paper in the current format.

Reviewer #2: Thank you for addressing my comments. I believe that the paper is now ready for publication

Reviewer #3: The authors have addressed my comments.

**Have the authors made all data and (if applicable) computational code underlying the findings in their manuscript fully available?**

Reviewer #1: Yes

Reviewer #2: Yes

Reviewer #3: Yes

PLOS authors have the option to publish the peer review history of their article (what does this mean?). If published, this will include your full peer review and any attached files.

Reviewer #1: **Yes: **PAVAN KUMAR S

Reviewer #2: **Yes: **Ghjuvan Grimaud

Reviewer #3: No

---

## [Editor Report · Acceptance letter]

PCOMPBIOL-D-25-01048R1

A compact model of *Escherichia coli* core and biosynthetic metabolism

Dear Dr He,

I am pleased to inform you that your manuscript has been formally accepted for publication in PLOS Computational Biology. Your manuscript is now with our production department and you will be notified of the publication date in due course.

With kind regards,

Zsofia Freund
